# Harnessing Epigenetics through Grafting: Revolutionizing Horticultural Crop Production

Qiang Jin [1,2,†], Muzafaruddin Chachar [3,†], Nazir Ahmed [4,†], Pingxian Zhang [5,6], Zaid Chachar [7], Yuke Geng [8,*], Dayong Guo [1,*] and Sadaruddin Chachar [9,*]

[1] National Key Laboratory for Germplasm Innovation & Utilization of Horticultural Crops, Huazhong Agricultural University, Wuhan 430070, China; qiangjin@taru.edu.cn
[2] College of Horticulture and Forestry, Tarim University, Alar 843300, China
[3] Department of Horticulture, Faculty of Crop Production, Sindh Agriculture University, Tandojam 70060, Pakistan; muzafar_mkc@yahoo.com
[4] Department of Crop Physiology, Faculty of Crop Production, Sindh Agriculture University, Tandojam 70060, Pakistan; nachachar@sau.edu.pk
[5] College of Life Sciences & Technology, Huazhong Agricultural University, Wuhan 430070, China; zhangpingxian@caas.cn
[6] Agricultural Genomics Institute at Shenzhen, Chinese Academy of Agricultural Sciences, Shenzhen 518000, China
[7] College of Agriculture and Biology, Zhongkai University of Agriculture and Engineering, Guangzhou 510550, China; zs.chachar@gmail.com
[8] College of Life and Environmental Sciences, Minzu University of China, Beijing 100081, China
[9] Department of Biotechnology, Faculty of Crop Production, Sindh Agriculture University, Tandojam 70060, Pakistan
* Correspondence: gengyuke@muc.edu.cn (Y.G.); guoday@mail.hzau.edu.cn (D.G.); schachar@sau.edu.pk (S.C.)
† These authors contributed equally to this work.

**Abstract:** Grafting is an ancient agricultural technique that is frequently used to enhance the performance of horticultural plants, including vegetables and woody fruit trees. For successful grafting, genotypes of the compatible scion (the upper part) and the rootstock (the lower part) must interact. Molecular signals, including nutritional and hormonal signals, proteins, and messenger RNAs (mRNAs), are known to be transferred from the rootstock to the scion and vice versa. Nonetheless, there are still numerous mysteries regarding artificial grafts, including the occurrence of genetic/epigenetic alterations due to exchanges between the graft partners, and the long-term ramifications of these alterations on the phenotype are unknown. Recent studies on the interactions between rootstocks and scions suggest that grafting responses have an epigenetic component. In this review, we focus on the current knowledge of epigenetic consequences following grafting. Epigenetic regulations are known to regulate chromatin architecture, alter gene expression, and affect cellular function in plants. Mobile small RNAs, for example, have been shown to modify the DNA methylation pattern of the recipient partner across the graft union. More recently, mRNA 5-methylcytosine (m$^5$C) modification has been shown to elucidate the long-distance transport mechanism of grafting in *Arabidopsis thaliana*. We also discuss how grafts can cause heritable epigenetic alterations that result in novel plant phenotypes, and how this might help increase horticultural crop quality, yield, and stress resistance in the context of climate change.

**Keywords:** grafting; epigenetic reprogramming; transcriptional regulation; DNA methylation; breeding; crop improvement

## 1. General Introduction to Grafting

Grafting is a centuries-old agricultural propagation approach frequently used to enhance the performance of plants in terms of production, quality, and resistance to biotic

and abiotic stresses. It entails fusing a rootstock with a scion, and two genetically distinct plant segments, so that the two sections combine and develop into a single plant. Grafting mostly depends on plants' inherent wound-healing ability [1,2]. This approach has been employed in agriculture for more than 2500 years. In modern agriculture, grafting has commonly been employed to produce a variety of horticultural crops, as well as forest trees [3,4]. Grafting is used to propagate uniform plantlets for commercial fruit species and to prevent juvenile states [5]. In addition, grafting can regulate plant growth, improve biotic and abiotic stress resistance, and enhance crop productivity and quality by forming scion-rootstock combinations [6,7].

Although the initial discovery of grafting is unclear, it most likely originates from the natural incidence of grafting, which occurs when two dissimilar plants accidentally come into contact and join their roots or branches without human involvement [8]. Artificial grafting by fusing the cut tissues of two different plants results in the establishment of a new plant with a single vascular system. Re-establishing the new plant entity begins with tissue connection at the grafting unions between the rootstock and scion, followed by a phase of rapid cell division, leading to the formation of a callus and a common cell wall, which is completed with the development of a distinctive vasculature system [9,10]. So far, a successful graft interaction may involve three main steps (Figure 1). (1) First, tissue adhesion between the rootstock and scion is initiated by the deposition of a homogeneous matrix containing pectins at the graft interface. (2) Second, cell divisions are then initiated, and a callus is formed (also known as pluripotent cells), connecting the scion with the rootstock. (3) Cell expansion is also observed and participates in gap filling; later, the pectin layer becomes thinner, and plasmodesmata form between the rootstock and scion cells. The phloem is the first to reconnect, followed by the resumption of root development, and finally, the xylem is connected. These grafting-related processes in *Arabidopsis thaliana* are rapid, requiring only one week to complete [10,11], but may need several months in woody plants [5]. The association between the two plant components is critical; even if the vascular connection between the two plant components has not yet been developed, the root responds transcriptionally quickly to the presence of the shoot [12].

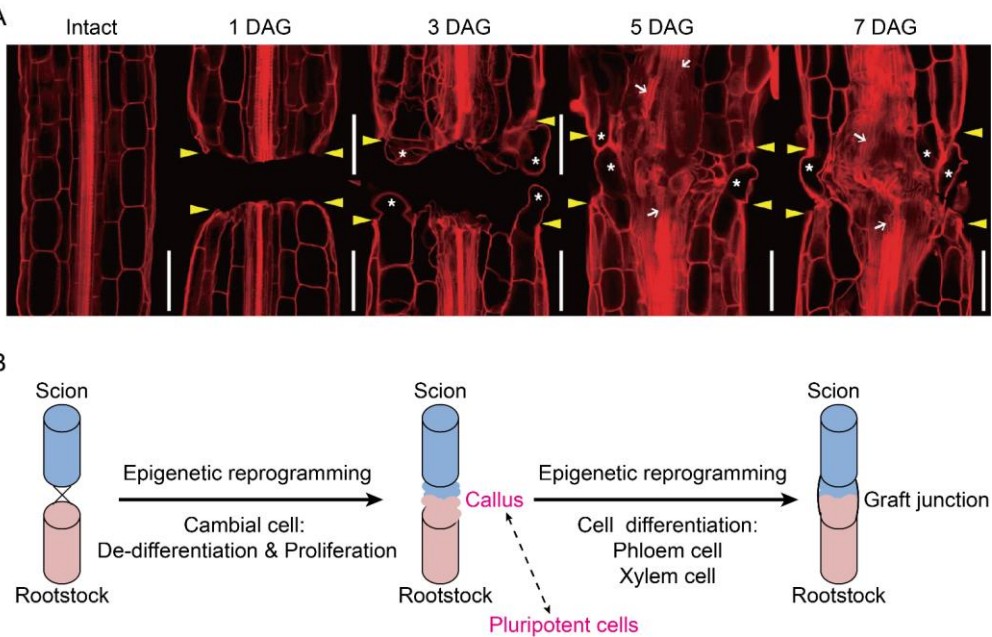

**Figure 1.** Morphological development and epigenetic reprogramming in the graft union. (**A**) Morphogenesis of the graft union using *Arabidopsis* micrografting as an example. Reprinted with permission

from Ref. [11]. Longitudinal sections were treated with modified pseudo-Schiff-propidium iodide (cell wall stain) and observed via confocal microscopy. From left to right: intact hypocotyl before cutting, and 1 day, 3 days, 5 days, and 7 days after grafting (DAG). (**B**) Epigenetic reprogramming during graft union development. Three main steps were involved in graft junction development, of which epigenetic reprogramming participates in cell de-differentiation, proliferation, and differentiation.

Grafting was considered to work best when performed between plants of the same family. Recently, it has been shown that plants from various families, even distantly related ones, can be grafted effectively [13]. Genes such as cellulases, which aid in cell wall repair, have been demonstrated to enhance grafting by promoting tissue adhesion [13]. If plants have these genes in abundance or are bred to have them, interfamily restrictions in grafting may be eliminated, and grafting techniques may be advanced to produce more chimeric plants that benefit from the advantages of both partners. During plant grafting, the rootstock, scion interaction, and molecular mechanisms regulating it are extremely complex (Figure 1) [11,14]. Discovering the molecular mechanism of grafting, with an emphasis on the interactions between the rootstock and scion in horticultural plants, is crucial. Although grafting is widely used, the molecular mechanisms driving it are still poorly understood despite recent advancements in models, woody plants, and vegetable species.

Plant grafting is thought to be mostly controlled by both epigenetic reprogramming and transcriptional regulation. The development of grafted plants and rootstock-scion interactions is thought to be mostly regulated by epigenetic modifications and transcriptional reprogramming. Epigenetics is the study of potentially heritable and stable modifications in the chromatin structure that have a substantial influence on gene expression and cellular function without altering the underlying DNA sequence. Three epigenetic modifications—DNA methylation, histone modification, and non-coding RNAs (ncRNAs), which are either small RNAs (sRNAs) or long non-coding RNAs (lncRNAs)) —are responsible for epigenetic alterations [15]. Epigenetic alterations affect numerous aspects of plant development, including fruit development and quality, yield, appropriate vegetative growth, efficient reproduction, and tolerance to environmental influences [15]. Epigenetics is associated with almost every aspect of plant development and the environmental factors affecting plants. Plant breeders can now apply genome-wide mapping of epigenetic marks and epigenetic targets to enhance and utilize epigenomic variability to obtain novel varieties of crops that are more resilient to climate change [16]. It is appropriate to determine whether plant grafting results in heritable epigenetic alterations that significantly affect gene expression variability and, consequently, the phenotypes of grafted plants.

The knowledge amassed to date about the genetic and epigenetic impacts associated with grafting, along with its potential for improving woody crops, are essential aspects to consider. The molecular and epigenetic aspects of grafting, however, remain poorly understood. In this review, we discuss current research and conclusions on the crucial epigenetic regulation of grafting processes, including tissue reconnection, growth of tissue vessels, the interaction between the rootstock and scion, genomic relationships, and the influence of rootstocks on scion performance, with special emphasis on the association between grafting and epigenetics.

## 2. Long-Distance Transport from Scion to Rootstock, or Vice Versa

Over the last two decades, the bidirectional transport of signal molecules over extended distances through the phloem of grafted and non-grafted plants has been extensively documented [12,17]. These significant discoveries have revealed that, in addition to nutritional and hormonal signals, other long-distance transport molecules such as mRNAs, proteins including RNA-binding proteins, and small RNAs integrate various intrinsic and external cues to orchestrate growth and development (Figure 2) [18]. The first instance of protein-coding RNA transfer across grafting partners was discovered in the Cucurbitaceae hetero-graft transcript *CmNACP,* involving *Cucurbita maxima* (pumpkin) [19]. The *CmNACP* gene encodes a NAC-domain transcription factor (TF) involved in meristem formation and

is crucial for controlling organ borders and organ development. The cucumber portion of the cucumber–pumpkin graft contained *CmNACP* messenger RNA (mRNA), indicating graft-induced transfer from the pumpkin rootstock to the cucumber scion through the phloem [19]. Similarly, an mRNA molecule encoding the phloem protein CmPP16, which was also found in stems, leaves, and floral scion tissues, migrated with its protein across the graft union and into the cucumber scion's phloem [20]. In pear, mRNAs containing the WUSCHEL-related homeobox (WOX) domain are transported from the scion to the rootstock through the phloem, assisted by the polypyrimidine tract-binding protein PbPTB3 and possibly regulating growth and flower development [21].

The enormous protracted transport of mRNA molecules occurs in a variety of homo-graft or hetero-graft systems. Using *Arabidopsis* ecotype-specific SNPs, it has been shown that thousands of mRNAs are transported across graft partners and that they respond to nutritional conditions in various ways [22]. Over 3000 transcripts migrated from cucumber scion leaves to watermelon sink tissues as an early response to phosphate deficiency stress [23], 138 transcripts migrated from the *Arabidopsis* rootstock to *N. benthamiana* scions, and 183 transcripts migrated from *N. benthamiana* scions to tomato roots [24]. These mobile mRNAs are thought to form a structure resembling tRNA, which provides stability, mobility, and translational ability [25]. Fascinating new research has demonstrated that cytosine methylation of mRNAs is necessary for the movement of mobile transcripts from shoots to roots in *Arabidopsis* grafts and that the translocation of a methylated *transnationally controlled tumor protein 1* (*TCTP1*) and *heat shock cognate protein 70.1* (*HSC70.1*) transcript to root tissue influenced root development [26]. Future research should elaborate on the previously addressed physiological importance of mobile transcriptomes between graft partners [25].

In addition to mobile mRNAs, plants have evolved in many mobile proteins, most of which are detected in the phloem (see in [18]). Remarkably, studies in *Arabidopsis*, tomato and rice (*Oryza sativa*) have identified the mobile flowering locus T (FT) protein as a 'florigen' or flowering hormone, which acts as a photoperiodic signal for seasonal fluctuations in day length to regulate flowering transition in long-distance transport (mobile) [27]. In potatoes (*Solanum tuberosum*), two different FT-like paralogues, StSP6A, that respond to independently regulated flowering time and tuberization are also involved in the long-distance transport from the leaf to the tuber (source-to-sink) [28]. Graft-induced modifications of several plant traits have been shown to produce numerous phenotypic variations, strongly suggesting that the grafted partners exchanged genetic information. Stegemann et al. discovered that tobacco transgenic lines containing several antibiotic markers and reporters produced cells that were antibiotic-resistant and showed both reporter genes at their graft sites. According to this study, genetic information is passed horizontally between the grafted partners, possibly as DNA fragments or plastids [29]. This study indicates that DNA fragments (genetic information) may be directly transported during graft union development.

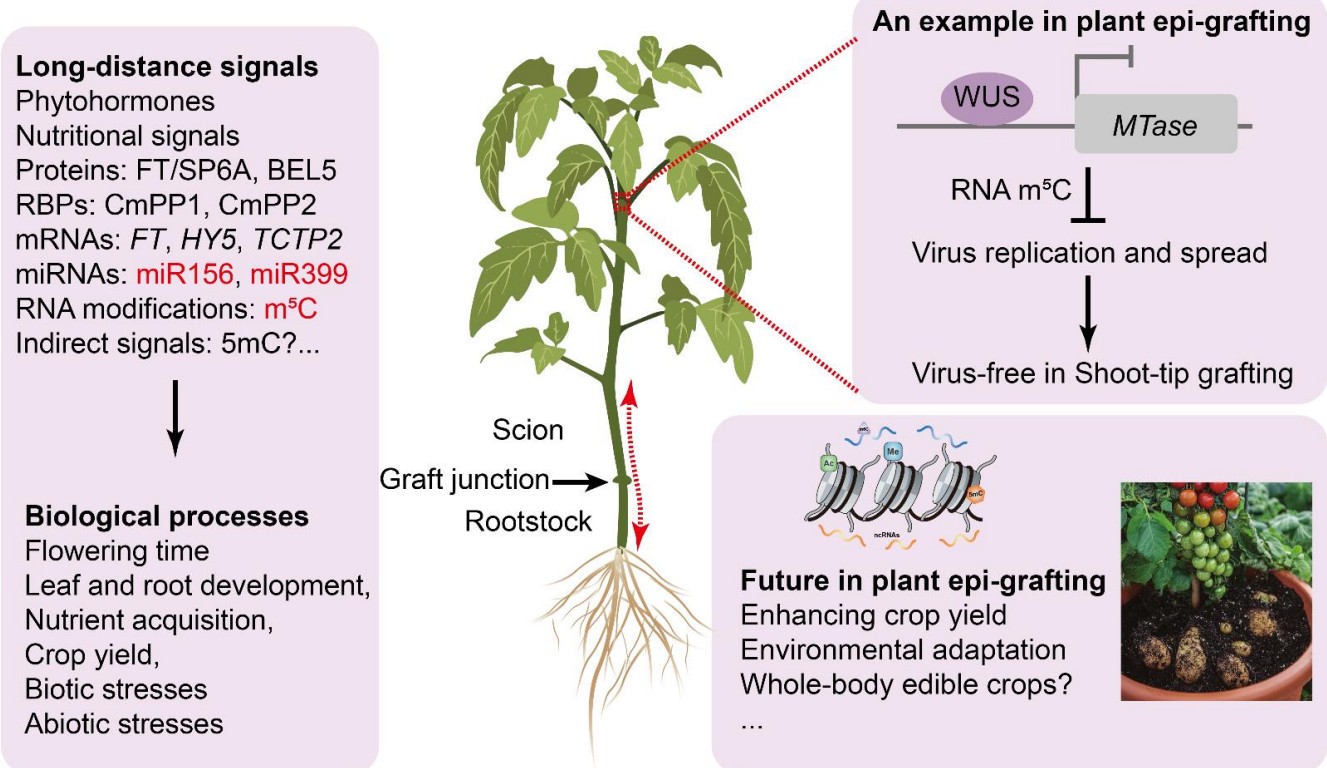

**Figure 2.** Long-distance-transport molecules and the model of epigenetic marks in plant grafting. In plants, many long-distance signals are essential in grafting. However, ncRNAs (miR156) and 5 methylcytosines (m⁵C) RNA can be transported over long distances (i.e., from scion to rootstock or the reverse). Thus, we assume that DNA methylation (and histone modifications) have an indirect impact on plant grafting via epigenetic reprogramming (See Figure 1B). Picture of 'TomTato' is adapted with permission from Ref. [30].

### 3. Small RNAs Play an Essential Role in Plant Grafting

Beyond mobile mRNAs and proteins, the observation that heritable epigenetic alterations identified in grafting partners may be caused by mobile sRNAs that travel through the phloem and grafting junctions, while following the source-to-sink gradient may increase the use of grafted plants in plant breeding. There are many mobile small RNAs associated with plant grafting (Table S1). For example, a study used miRNAs miR156 and miR172 associated with juvenility and flowering and their probable target genes to evaluate pre-graft and post-graft material from avocado in various combinations, confirming the graft transmissible regulation of miR156 and miR172 [31]. The origin, transport, and function of these small RNAs in epigenetic alterations have been largely understood, but how they participate in the epigenetic reprogramming of plant grafting, in connection with grafting plasticity, is lacking.

Small RNA expression varied among the three progeny selfing generations following grafting. Compared to the leaves of non-chimeric plants, the expression of several relevant 24-nt siRNAs was reduced. These siRNAs were linked to regions with differential methylation. In another investigation, SAM termination was not constant, whereas the leaf form was once again found to be stable. In addition, compared to non-chimeric controls, several micro-RNAs (miRNAs) displayed unique expression patterns in the reverted progenies, and these expression patterns were accompanied by modifications in the expression of their target genes [32]. Intra-species/inter-cultivar grafting was employed in *Cucurbita pepo* to examine the impact of grafting on the quality of scion fruit, and methylation and miRNA analyses were carried out to track the fruit phenotypic changes that occurred after grafting. A previous study has performed homo-grafting as well as reciprocal grafting

of cucumber and pumpkin [33]. After sequencing the RNA from the leaves and root tips of the grafted plants, they assessed the expression of miRNAs in hetero-grafts and homo-grafts tissues. When compared to homo-grafted tissue, they discovered that most of the expression of miRNAs was altered in hetero-grafts [33]. However, miRNA expression in grafted plants cannot be examined independently of the stress. In homo- and hetero-grafted cucumbers grafted onto pumpkin rootstocks under salt stress, distinct miRNA expression patterns were discovered, demonstrating that salt stress adaptation may affect miRNA regulation [34]. It remains unclear whether various rootstock and scion combinations directly contribute to these effects or whether stress adaptations are caused by miRNAs or their effects.

Moving epigenetic signals may have significant effects on species, such as cucumber and pepper that often undergo grafting. Small RNA mobility in other species of the Solanaceae family was established using transgenic *Nicotiana benthamiana* scions that generate siRNAs grafted on potato rootstocks that either express or do not express Green Fluorescent Protein (GFP). In conjunction with hypermethylation of the targeted area, small RNAs have been shown to silence GFP transcription in the potato lateral roots [35]. Regenerated microtubers and shoots produced from root tissue cultures showed TGS of GFP. The *endogenous granule-bound starch synthase I (GBSSI)* gene target region on two micro-tubers of the adventitious shoots produced by grafting *N. benthamiana* transgenic scions onto WT potato rootstocks had significant levels of methylation. Both the shoots produced by these micro-tubers and the second offspring tubers in the absence of sRNA maintained an elevated methylation status, demonstrating that the trait was stably inherited [35].

In vegetable rootstocks and scions and in grafts of woody plants, both protein-coding and non-coding sRNAs induced by grafting have been reported [36,37]. These graft-induced transcripts may function in graft development, fruit quality, scion yield, and biotic and abiotic stress responses. Grafting has been used in several studies to demonstrate that mRNA molecules move through plants in the phloem. sRNAs have also been shown to be involved in grafting. The phenotypic variations caused by grafting appear to be mostly influenced by transcriptional and epigenetic modifications [8]. Several plant species have been shown to transfer small RNAs to grafted plants [38]. According to published research, mobile sRNAs can silence genes in graft partners by degrading the corresponding mRNA target molecules or by RNA-directed DNA methylation of targeted genomic loci [38,39]. Transgene-derived and endogenous sRNAs were used to demonstrate the capability of 24-nt siRNAs to direct DNA methylation at three positions along the graft union in the recipient cell genome [38]. Mobile siRNA signals that migrated to the roots of an *Arabidopsis* graft governed extensive methylation processes in the recipient root genome [40].

Small RNAs, which carry genetic information, have received considerable attention in recent years. The small RNA miR399, which is generated by phosphorus deprivation, was the first to be found in the phloem sap of rapeseed and pumpkin, and micro-grafting tests showed that it migrates from the shoots to the roots [41]. Furthermore, miRNAs have been suggested to be transferred from the scion to the rootstock, including miR156, miR172, miR395, and miR399 [42,43]. However, fundamental studies in *A. thaliana* have recently enabled a new understanding of genetic information moving across grafted plants in the form of siRNAs. In RdDM, primarily heterochromatic 24-nt siRNAs drive the de novo methylation of predominantly transposable elements (TEs) and repetitive DNA [44], resulting in TGS.

Molnar et al. demonstrated that many sRNAs relocate inside grafted plants, but predominantly 24-nt siRNAs generated by the action of a Dicer-like protein, DCL3, are mobile and transmitted to the roots from the shoots [38]. This study was performed using WT and mutant *Arabidopsis* plants grafted onto one another. Although the other classes of siRNA, such as 22-nt and 23-nt, are equally mobile, 24-nt siRNAs were linked to DNA methylation at three specific loci, all TEs in the roots [38]. The phloem is responsible for this siRNA transport. Subsequently, the ability of only mobile 24-nt siRNAs to direct DNA methylation in recipient meristematic tissues and the TGS of a transgene promoter was

further demonstrated by using transgenic *Arabidopsis* plants [24]. Additionally, despite not being the only class of mobile sRNAs, mobile 24-nt siRNAs have been demonstrated to migrate from the shoot to the root and could direct DNA methylation of thousands of genomic regions that are connected with TEs [40].

As sRNAs are thought to migrate through the phloem and plasmodesmata, their movement in grafted plants is more effective when they are created in the scion and move towards the rootstock rather than vice versa. Nevertheless, siRNAs generated in rootstock phloem-associated cells go to the WT scion and diminish the level of viroid infection assisted by the loss of buds and lateral leaves [45]. Non-transgenic scions were grafted onto transgenic tomato rootstocks with a silenced fatty acid desaturase gene (*LeFAD7*). *LeFAD7* gene expression was decreased in grafted scions, and siRNAs were found, indicating that the transgenic rootstock had transported them to the scion. Additionally, the leaves of scions in this study were removed prior to grafting [46]. The process of mentor grafting involves the removal of leaves from the scions to enhance the transportation of substances from the rootstock to the scion. It is probable that, among these substances, sRNAs produced from the rootstock will also move in that direction, emphasizing the considerable phenotypic alterations observed in the scions.

There may be many practical ramifications for the apparent migration of sRNAs inside the grafted plants. For instance, in tomato plants, viral resistance is passed from the rootstock to the scion. Spanò et al. produced TSWV (tomato spotted wilt virus)-resistant scions that accumulated less viral RNA when they were grafted onto a tomato variety resistant to TSWV, which has a higher RNA interference (RNAi) response to the viral infection. Using qRT-PCR, it was demonstrated that resistant grafted plants showed upregulation in the expression of major RNAi mechanism genes such as Argonaute (AGO) and RNA-dependent RNA polymerase (RDR) in their roots. The discovery that RNA silencing was more pronounced in self-grafted plants demonstrates that even grafting may cause the mechanism to become activated [47].

Additionally, if one of the grafting partners is GM, researchers may be able to use a new plant breeding technique to avoid GM concerns. The flowers or fruits of the other grafting partner, commonly the scion, will not be termed GM because the output originates from the non-GM plant component, which obviously benefits from the genetic modification advantages of the other grafting partner. This novel approach, also known as "transgrafting", has previously been reported and is based on the mobility of distinct genetic materials that may migrate within grafted plants [48]. Agriculture and plant sciences should consider the possible use of grafting resistant rootstock or even practicing trangrafting to produce crops with improved quality and higher yield, considering recent concerns over GM plants and their regulations and in light of Europe's cautious attitude towards GM technology and products.

## 4. DNA Methylations Have an Indirect Impact during Graft-union Development

Plants have three fundamental epigenetic processes that can activate or silence their genes. An increasing amount of research indicates that, in addition to small RNA signals, other types of epigenetic marks, such as DNA methylations, cause a variety of phenotypic changes in grafted plants. Grafting-related epigenetic alterations have been associated with small RNAs and DNA methylation. For example, scion siRNAs have been shown to cause RNA-directed DNA methylation (RdDM) in the rootstock [49].

Grafted plants often exhibit altered DNA methylation. Although one would counter that this is not surprising given the strong evidence linking changes in DNA methylation to stresses, such as cutting and wounding, it seems that there is more than just wound stress involved in the association between DNA methylation and plant grafting. In grafted Solanaceae plants, tomatoes and eggplants were grafted onto one another, but pepper was only utilized as a rootstock for tomatoes in a study conducted by Wu et al. (2013) to demonstrate alterations in DNA methylation. The overall methylation levels of grafted plants did not change, according to the analysis of methylation-specific amplified markers

(MSAP) [50]. In scions and pepper rootstocks, only locus-specific modifications to local methylation were detected. A high proportion of these DNA methylation alterations in scions was transmitted to self-pollinated graft progenies. Additionally, using bisulfite sequencing (BS-seq) of specific loci, it was discovered that even self-grafting can cause some DNA methylation alterations. Significantly heritable cytosine methylation modifications have been found to be associated with interspecies grafting in Solanaceae.

Some DNA methylation-related genes, such as *Methyl Transferase (MET) 1*, showed drastically modified expression profiles in tomato/eggplant grafted plants, albeit these profiles were reversed in progenies compared to their seed non-grafted controls [50]. A study by Wu et al. showed that grafting produces heritable changes in DNA methylation and that the observed variation may be due to altered gene expression of epigenetic regulators such as met1. This was one of the first studies to explicitly link DNA methylation and epigenetics to grafting, particularly in scions. Another investigation of interspecies grafting in the Cucurbitaceae plant family also suggested that DNA methylation may be involved in the ramifications of grafting in other plant families. Researchers grafted cucumber, melon, and watermelon onto a pumpkin and used methylation-sensitive amplification polymorphism (MASP) markers to detect changes in global DNA methylation. Interestingly, they found a significant increase in global DNA methylation in cucumber and melon scions, but an intriguing lack of change in watermelon [51]. This suggests that the interaction between the rootstock and the scion may be the exclusive source of epigenetic alterations in grafting scions.

Numerous investigations on model species have shown that the long-distance bidirectional movement of macromolecules, such as mRNA transcripts, miRNAs, and other sRNAs, is a potential biological process implicated in grafting. These RNAs can induce physiological changes through the graft junction that may lead to the development of vigor. Several plant species exhibit considerable DNA methylation alterations induced by grafting. Moreover, utilizing interspecific grafts, locus-specific alterations in DNA methylation in eggplant, tomato, and pepper scions have been documented and passed down to progeny [50]. Although the change in this instance was transgenerational, it was also reversible after several generations. Moreover, numerous studies have demonstrated that grafting causes extensive transcriptome alterations in plants. Unfortunately, there is insufficient research at the epigenetic level to draw broad conclusions. In addition, most studies to date have concentrated on variations in DNA methylation triggered by grafting; however, there have been no studies on alterations in histone modifications caused by grafting. Understanding the process behind grafting-mediated alterations in gene expression requires future research into the potential function of histone epigenetic marks established by grafting.

However, in an investigation of grafting in Solanaceae using *Solanum esculentum*, *S. melongena*, and *Piper nigrum*, global DNA methylation levels were substantially altered, inducing hypomethylation and hypermethylation in numerous loci induced by grafting. These epigenetic alterations were passed down to the self-pollinated progeny [50]. The scions of *Cucumis melo* and *C. sativus* grafted onto *C. pepo* rootstocks were also shown to have DNA hypermethylation in inter-species hetero-grafting [51]. Finally, the rubber tree (*Hevea brasiliensis*) scion was found to have DNA methylation polymorphisms, including DNA hypermethylation of the promoter and the coding sequence of the LRR receptor kinase gene [52]. Taken together, these findings imply that grafting can be used to improve vegetables and other crops by altering their epigenome.

Studies on citrus grafts have demonstrated differential DNA methylation levels in different scion/rootstock combinations [53,54]. In another investigation on *H. brasiliensis*, methylation variations were discovered across buds from a single seedling produced from polyembryony and grafted onto a genetically distinct rootstock [52]. Grafting studies in *Arabidopsis* have demonstrated that long-distance transport of endogenous and transgene-derived sRNAs through the graft junction may compel DNA methylation in the genome of recipient cells, thereby triggering physiological modifications [38]. In eggplant (*Solanum*

*melongena*), CMT3b and MET methyltransferases were shown to be upregulated in hetero-grafted scions [55].

In *Arabidopsis* and tomato, plant vigor generated by impaired mitochondrial- and plastid-targeted protein (MSH1) function may be transferred across a graft union, which has also been implicated in epigenetic regulation [56]. Methylation in the CHH domain is generally related to a suppressed TE expression in plants. As a result, the observed genome-wide reduction in methylation in hetero-grafted scions was not associated with a broad increase in TE expression, but rather with a more nuanced modulation causing the downregulation of numerous TEs. In a recent study on epigenetic mutants of tomatoes, modulation of LTR-TEs occurred in an age-dependent manner [57]. This modulation was linked to the preferential DNA methylation of young transposons by the DNA methyltransferase CMT3 [57].

In hetero-grafted plants, the upregulation of DNA methylation-related genes might be linked to the suppression of gene expression or the activity of transposons. Several histones being upregulated may be connected to the chromatin structure alterations of hetero-grafted plants and may be implicated in some of the discrepancies in gene expressions reported between auto- and hetero-grafted plants.

## 5. RNA Modification Acts as an Additional Player in the Regulation of Plant Grafting

While mRNA 5-methylcytosine (m$^5$C) mRNA alteration in horticultural plants has not yet been documented, its significance in *Arabidopsis* has drawn the attention of researchers interested in understanding the grafting process [26,58]. Nutrients, proteins, hormones, and genetic materials are transported through the graft junctions. In addition to other epigenetic alterations, m$^5$C alterations in thousands of mRNAs are transported to different plant regions via graft union through the phloem. In *Arabidopsis*, m$^5$C alteration of mRNAs assists their transport, and mobile m$^5$C-modified *TCTP1* and *HSC70.1* were found to modify root development by translocating into target cells [26]. Another recent study in *Arabidopsis* showed that m$^5$C-mediated *WUSCHEL* (*WUS*) stimulates innate antiviral immunity [59]. With the use of shoot-tip grafting (STG) techniques, it has become increasingly important to develop virus-free plants to combat various viral diseases in many horticultural species [60]. Realizing how mRNA m$^5$C directs grafting would be beneficial in horticultural plants, given that the majority of horticultural crops are extensively propagated through grafting.

## 6. Conclusions and Future Perspectives

In conclusion, this review provides insights into the epigenetic consequences of grafting in horticultural crops. Recent findings underscore the significance of understanding the epigenetic response caused by grafting, as it has the potential to greatly impact plant phenotypes, horticultural crop quality, yield, and stress resistance. Understanding the epigenetic consequences of grafting opens up exciting avenues for improving horticultural practices. By harnessing the potential of graft-induced heritable epigenetic alterations, we can explore strategies to enhance plant performance and adaptability in the face of environmental challenges, including the impacts of climate change. These alterations offer promising opportunities to increase horticultural crop quality, yield, and stress resistance, thereby contributing to sustainable and resilient agriculture.

Interactions in grafting are intricate. Recent investigations have shown that grafting modifies gene expression, which influences the phenotype of the scion. These studies also suggested that the performance of grafted plants is frequently and significantly influenced by the interaction of genotypes. In grafted plants, it appears that this interaction creates unique transcriptome patterns that frequently result in the complete reprogramming of gene expression. However, how the grafting process affects gene expression cannot be ignored. Future research on the molecular processes that regulate grafting and its connected interactions must address these factors independently in each grafting experiment while employing necessary controls and a comprehensive analysis of RNA-seq data.

High-throughput sequencing has made it possible to discover and access many reference genomes for a variety of plant species. It is now feasible to acquire a better understanding of the genomic interactions that occur during grafting using advanced molecular methods and extensive information present in bioinformatics databases. In addition, it is now more practical to clarify the molecular mechanisms that support the establishment of grafting as well as the movement of genetic material inside grafted plants. The increased usage of grafting in additional plant species, which contributes to biodiversity conservation and sustainability, may be facilitated by resolving compatibility concerns. To fully understand the influence on the phenotype and function of grafted plants, as well as to guide technology towards making the most of the malleable epigenome of grafted plants, further research must be conducted. Epigenetic varieties could serve as new sources of phenotypic diversity, enabling adaptation to changing environments [61].

As a result of the mapping of epigenetic modifications in plant genomes and the discovery of epigenetic targets, breeders may have new techniques to improve and exploit epigenetic variability in their attempts to breed new crop varieties. Cell–cell adhesion is a viable target for improving the plant grafting processes. It has recently been demonstrated that overexpressing b-1,4-glucanases promotes grafting, since at the graft interface, a subclade of b-1,4-glucanases produced in the extracellular region aids in cell wall reconstruction. In one study, tomato fruit was produced from other plant families using *Nicotiana* stems as an interscion [13], demonstrating that two incompatible species can be grafted. This finding demonstrates that grafting incompatibility can be overcome, enabling grafting beyond closely related species [30]. When we obtain a deeper understanding of plant grafting, we may be able to use all the opportunities it presents or even manage it to our benefit for agricultural output.

Recently, embryonic hypocotyl has been shown to allow inter- and intra-specific grafting in all three groups of monocotyledons: commelinids, lilioids, and alismatids. Nevertheless, it was long believed that since monocotyledons lack vascular cambium, grafting would not be possible in this second-largest group of terrestrial plants, which includes many staple crops [62]. Further research on the potential of grafting in monocotyledons is required to fully understand the implications of this discovery. This breakthrough will lead to new and innovative ways to improve crop productivity and sustainability and explore the potential of monocotyledons to create new and improved crop varieties through grafting. Another recent investigation has achieved heritable gene editing by fusing Cas9 and guiding RNA transcript-like sequence motifs that transport RNAs to WT shoots (scions) from transgenic rootstock through grafting. The graft-mobile gene editing approach facilitates the transgene-free production of offspring in one generation without requiring transgene removal, culture recovery, and selection, or the use of viral editing vectors [63]. This will reduce concerns around the potential environmental and health risks associated with transgenic crops, considering recent concerns over GM plants and their regulations and considering Europe's cautious attitude towards GM technology and products.

However, there are still many unanswered questions in this field that warrant further investigation. Future research should aim to unravel the specific mechanisms underlying epigenetic modifications during grafting, explore the diversity of epigenetic alterations induced by grafting in different plant species, and elucidate the transgenerational stability and inheritance of these alterations. Furthermore, it will be valuable to investigate the potential interactions between epigenetic and genetic factors in determining the phenotypic outcomes of grafted plants. Future research should also focus on unraveling the molecular processes and genomic interactions underlying grafting-induced changes in gene expression, exploiting epigenetic variability for crop breeding, optimizing grafting techniques for diverse plant species, exploring grafting in monocotyledons, and further advancing gene editing technologies via grafting. These avenues of investigation hold immense potential to revolutionize horticultural practices, enhance crop productivity, and contribute to sustainable agriculture in the face of global challenges.

The epigenetic regulation of grafting represents a promising area of research with significant implications for horticulture. By deciphering the complex interplay between epigenetic modifications and grafting responses, we can unlock the full potential of this ancient agricultural technique to enhance crop performance and address the challenges posed by a changing climate. Continued exploration in this field will undoubtedly contribute to the advancement of sustainable and resilient horticultural practices.

**Supplementary Materials:** The following supporting information can be downloaded at: https://www.mdpi.com/article/10.3390/horticulturae9060672/s1, Table S1: List of some miRNAs in the regulation of plant grafting [31,33,37,42,43,64–68].

**Author Contributions:** Conceptualization, P.Z.; writing—original draft, S.C., Q.J., N.A. and M.C.; supervision, S.C., Y.G. and D.G.; funding acquisition, P.Z.; writing—review and editing, S.C., P.Z. and Z.C. All authors have read and agreed to the published version of the manuscript.

**Funding:** This research was funded by the Natural Science Foundation of Hubei Province of China (2022CFB638), the National Key R&D Program of China (2019YFD1000600), the China Postdoctoral Science Foundation (2022M711268), and Leading program of Young academic team in Minzu University of China (No. 02200201).

**Institutional Review Board Statement:** Not applicable.

**Informed Consent Statement:** Not applicable.

**Data Availability Statement:** Not applicable.

**Acknowledgments:** We apologize to colleagues whose work could not be cited due to lack of space.

**Conflicts of Interest:** The authors declare no conflict of interest.

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
