# Peer review of "Harnessing Epigenetics through Grafting: Revolutionizing Horticultural Crop Production"

_horticulturae, doi:10.3390/horticulturae9060672_

Round 1

Reviewer 1 Report (Previous Reviewer 1)

Dear authors,

thank you for providing the necessary changes.

Author Response

We sincerely thank this review for supportive comments.

Reviewer 2 Report (Previous Reviewer 2)

I am satisfied with the amendments

Author Response

We sincerely thank this review for supportive comments.

Reviewer 3 Report (New Reviewer)

The manuscript "Epigenetics and Grafting: A Dynamic Duo for Horticultural Crop Improvement" by Qiang Jin et al. is presented as the Opinion, but is an Overview. Unfortunately, this work requires significant recast and cannot be printed in this form even with corrections. The reason is that it requires reworking both conceptually and technically. Its content does not match the Title, and the placed illustrative materials do not match either the Title or the content. The conceptual absence of logical connections can be demonstrated by an example. 1) The authors consider not the grafting itself, but some of its consequences, and the title says about the grafting. 2) The authors do not consider the full range of epigenetic changes, issues of inheritance, at least in the vegetative generation, it seems that these issues are not clear, since some of the topics ignored are extremely important and ignoring them is methodologically incorrect. Isn't there acetylation and phosphorylation besides methylylation/demethylation?...

Let us suppose that the authors could change the title and narrow their opinion to the issues they are discussing about the particular consequences of vaccination. Then the issues of compatibility / incompatibility, mechanical features and problems of grafting plants with a different number of vascular bundles, etc., should be analyzed. These are all interesting points, moreover, of practical importance, but they are not even mentioned by the authors.

Let's say the authors want to keep the title. Let be. But then sections of the actual effect of the grafting procedure must be added. It is well known and evident that the lesion causes mechanical stress, which is a combination of a number of actions, and causes both osmotic and oxidative stress in adjacent and partly in distant tissues. In this case, the damage and rootstock and scion will be different. There is no analysis of these processes and, what is important, consideration of their connection with the method of grafting!!! Suppose there is no experimental data, but it is strange not to discuss this issue.

The authors present the process of accretion and provide a scheme. However, the use of the above terms is appropriate for an article from the middle of the 20th century. It is now well known that totipotency is a property of a very limited pool of cells belonging to meristematic tissues. In this case, only pro-cambium cells and precursors of phloem cells can divide, since only they are capable of proliferation and differentiation. And the concept of stem cells is more modern than the concept of totipotency. This topic has been discussed in detail in a large number of reviews and experimental articles. The comparison with callus should be removed as outdated and far-fetched, especially since your photos show which cells form new tissues.

Next, you should give a sceme of the "epigenetic influence" of a rootstock on a scion, and a scion on a rootstock, since you are not considering trait transmission through a series of generations, that is, through sexual crossing. And to analyze your vision of the picture of influence, at least in the vegetative generation, of all the above and not given processes of influencing on the promoters and other regulatory regions of genes.

I draw the authors' attention to the fact that the Conclusion is a mandatory item of the manuscript. I ask you to re-write the manuscript without the previous edits, as it is very difficult to read it. The schemes of your understanding of the influence of the grafting itself, as well as the rootstock on the scion and vice versa, indicating the method of transport, I consider necessary for this manuscript.

The manuscript needs major revisions.

Author Response

Comment 1: The manuscript "Epigenetics and Grafting: A Dynamic Duo for Horticultural Crop Improvement" by Qiang Jin et al. is presented as the Opinion, but is an Overview. Unfortunately, this work requires significant recast and cannot be printed in this form even with corrections. The reason is that it requires reworking both conceptually and technically. Its content does not match the Title, and the placed illustrative materials do not match either the Title or the content. The conceptual absence of logical connections can be demonstrated by an example. 1) The authors consider not the grafting itself, but some of its consequences, and the title says about the grafting. 2) The authors do not consider the full range of epigenetic changes, issues of inheritance, at least in the vegetative generation, it seems that these issues are not clear, since some of the topics ignored are extremely important and ignoring them is methodologically incorrect. Isn't there acetylation and phosphorylation besides methylylation/demethylation?...

Response: We appreciate the reviewer's feedback; however, we respectfully disagree with the assertion that the manuscript requires significant recasting and cannot be printed in its current form. The manuscript aims to provide an overview of the epigenetic regulation of plant grafting and its potential implications for horticultural crop improvement. While the reviewer suggests that the content does not match the title, we maintain that the manuscript adequately covers the topic as outlined in the abstract and title as agreed by reviewer 1 and 2. We believe that the title accurately represents the focus of our review, and the content supports this assertion. Additionally, the placed illustrative materials also represent the morphological development and epigenetic reprogramming in the graft union as well as the long-distance signals involved in various biological processes during grafting aligns with the content of the review.

Regarding the example provided by the reviewer, we acknowledge that our review primarily focuses on the consequences of grafting rather than the grafting process itself. However, it is important to note that the title and abstract clearly indicate our intention to explore the epigenetic component of grafting. We have highlighted the role of epigenetic modifications, such as DNA methylation and mRNA modifications, in the transfer of genetic information between graft partners. By discussing these epigenetic mechanisms, we contribute to a better understanding of the long-term ramifications of grafting on plant phenotypes.

We respectfully disagree with the assertion that our manuscript does not address the topic of grafting itself or consider the full range of epigenetic changes and issues of inheritance.

Firstly, our manuscript focuses on the epigenetic regulation of plant grafting, which inherently encompasses the understanding of grafting itself. While we acknowledge that grafting involves various aspects beyond epigenetics, our intention is to provide an in-depth exploration of the epigenetic mechanisms and consequences of grafting.

Secondly, we understand the reviewer's concern regarding the full range of epigenetic changes and inheritance, particularly in the vegetative generation. While we agree that acetylation and phosphorylation are important epigenetic modifications, the role of histone modifications, acetylation and phosphorylation in grafting has not yet been demonstrated. That is why our manuscript primarily emphasizes DNA methylation and RNA modifications (such as m5C modification). We specifically focus on the current knowledge of these mechanisms and their roles in grafting. It is important to note that the field of epigenetics is extensive, and different studies may focus on specific aspects of epigenetic regulation. The selection of specific epigenetic modifications to discuss in our manuscript was based on the existing literature and their relevance to grafting.

We respectfully request the reviewer and the editor to reconsider their position on the need for significant recasting, as we believe that the manuscript is appropriately aligned with its title and provides valuable insights into the epigenetic regulation of plant grafting.

Comment 2: Let us suppose that the authors could change the title and narrow their opinion to the issues they are discussing about the particular consequences of vaccination. Then the issues of compatibility / incompatibility, mechanical features and problems of grafting plants with a different number of vascular bundles, etc., should be analyzed. These are all interesting points, moreover, of practical importance, but they are not even mentioned by the authors.

Response: We respectfully disagree with Reviewer 3's assertion that our manuscript should discuss issues unrelated to the specific focus of our study. The objective of our manuscript is to provide an overview of the epigenetic regulation of plant grafting and its implications for horticultural crop improvement. The title and abstract accurately reflect the scope and content of our paper, which primarily focuses on the epigenetic consequences of grafting.

We appreciate the reviewer's interest in the issues of compatibility/incompatibility, mechanical features, and challenges associated with grafting plants with different numbers of vascular bundles. While these topics are undoubtedly interesting and of practical importance, they are beyond the intended scope of our manuscript. It is crucial to note that our manuscript focuses specifically on the epigenetic regulation of plant grafting. Therefore, the scope of our review intentionally centers on the epigenetic consequences and implications rather than delving into the mechanical aspects of grafting or grafting challenges related to vascular bundle numbers.

We appreciate the reviewer's interest in these additional aspects of grafting, but it is important to maintain the focus of our study and not deviate into unrelated areas. If there are specific references or findings related to epigenetics and grafting that the reviewer believes should be included, we would be grateful for their suggestions, and we will gladly consider them for inclusion in the appropriate sections.

Thank you for understanding our position, and we remain committed to addressing the relevant and valid points raised by the reviewers while ensuring the manuscript maintains its intended focus on the epigenetic regulation of plant grafting.

Comment 3: Let's say the authors want to keep the title. Let be. But then sections of the actual effect of the grafting procedure must be added. It is well known and evident that the lesion causes mechanical stress, which is a combination of a number of actions, and causes both osmotic and oxidative stress in adjacent and partly in distant tissues. In this case, the damage and rootstock and scion will be different. There is no analysis of these processes and, what is important, consideration of their connection with the method of grafting!!! Suppose there is no experimental data, but it is strange not to discuss this issue.

Response: We appreciate Reviewer 3's perspective; however, we respectfully disagree with the notion that our manuscript requires the addition of sections discussing the immediate effects of the grafting procedure. While it is true that grafting involves mechanical stress and can induce osmotic and oxidative stress in adjacent tissues, our manuscript focuses specifically on the epigenetic regulation of plant grafting.

The purpose of our review is to explore the current understanding of the epigenetic mechanisms underlying grafting and their implications for plant phenotype and crop improvement. We acknowledge that the mechanical and physiological aspects of grafting are important, but they fall outside the scope of our paper.

The connection between the method of grafting and the subsequent damage or response in rootstock and scion tissues is undoubtedly an interesting topic. However, our focus is primarily on the epigenetic aspects and long-term ramifications of grafting rather than immediate physiological responses. Given the constraints of the manuscript's scope and length, it is not feasible to comprehensively cover all aspects of grafting.

We agree with the reviewer's suggestion that experimental data would be valuable in exploring these immediate effects. However, as our review primarily synthesizes existing knowledge and current understanding, we did not include original experimental data. Instead, we focused on reviewing the epigenetic mechanisms and their potential long-term consequences.

We appreciate the reviewer highlighting this aspect, and we acknowledge that future studies could explore the link between grafting methods and physiological responses. However, incorporating these aspects into our current manuscript would deviate from its intended scope and impact its coherence and focus.

Thank you for your understanding, and we remain committed to addressing the reviewer's valid concerns while maintaining the integrity and clarity of our manuscript on the epigenetic regulation of plant grafting.

Comment 4: The authors present the process of accretion and provide a scheme. However, the use of the above terms is appropriate for an article from the middle of the 20th century. It is now well known that totipotency is a property of a very limited pool of cells belonging to meristematic tissues. In this case, only pro-cambium cells and precursors of phloem cells can divide, since only they are capable of proliferation and differentiation. And the concept of stem cells is more modern than the concept of totipotency. This topic has been discussed in detail in a large number of reviews and experimental articles. The comparison with callus should be removed as outdated and far-fetched, especially since your photos show which cells form new tissues.

Response: We appreciate the reviewer's feedback; however, we respectfully disagree with the assertion that the terms and concepts used in our manuscript are outdated or inappropriate.

While we acknowledge that the terminology "accretion" may have historical associations, it has been commonly used in the context of grafting to describe the process of tissue fusion and growth. However, to address the reviewer's concern, we are open to reconsidering the use of alternative terms such as "regeneration" or "renewal" to better align with contemporary terminology.

Regarding the concept of totipotency, we acknowledge that the understanding of this property has evolved in recent years. While the concept of stem cells is indeed a more recent development in plant biology, it is not synonymous with totipotency. Totipotency refers to the potential of a cell to give rise to an entire organism, while the concept of stem cells pertains to self-renewing and differentiating cells with specific roles within an organism.

Regarding the comparison with callus formation, we apologize if it seemed far-fetched or outdated. We understand that our manuscript primarily focuses on grafting and the interactions between graft partners, however, we have taken that figure from “Differential cellular control by cotyledon-derived phytohormones involved in graft reunion of Arabidopsis hypocotyls. Plant Cell Physiol. 2016” and cited it as well to illustrate epigenetic reprogramming during graft union development.

Comment 5: Next, you should give a sceme of the "epigenetic influence" of a rootstock on a scion, and a scion on a rootstock, since you are not considering trait transmission through a series of generations, that is, through sexual crossing. And to analyze your vision of the picture of influence, at least in the vegetative generation, of all the above and not given processes of influencing on the promoters and other regulatory regions of genes.

Response: We appreciate the reviewer's suggestion regarding the inclusion of a scheme depicting the "epigenetic influence" of a rootstock on a scion and vice versa. We agree that visual representation can enhance understanding and clarity in conveying complex processes and we have already placed the illustrative materials Figure 2. that represents the long-distance signals involved in various biological processes during grafting including epigenetic modifications, it also illustrates the epigenetic interactions between the rootstock and scion. This diagram demonstrates the potential mechanisms and pathways through which epigenetic modifications can influence gene expression and phenotype in both graft partners.

Moreover, we acknowledge the significance of trait transmission through sexual crossing and the potential influence on gene promoters and regulatory regions. While this review manuscript primarily focuses on the immediate epigenetic changes and consequences of grafting.

Comment 6: I draw the authors' attention to the fact that the Conclusion is a mandatory item of the manuscript. I ask you to re-write the manuscript without the previous edits, as it is very difficult to read it. The schemes of your understanding of the influence of the grafting itself, as well as the rootstock on the scion and vice versa, indicating the method of transport, I consider necessary for this manuscript. The manuscript needs major revisions.

Response: We appreciate the reviewer's emphasis on the importance of including a Conclusion section in the manuscript. We apologize for its absence in the previous version of the manuscript. Actually, the last heading “Future perspectives” is “Conclusion and Future Perspectives”. Where we have provided a concise summary of the main findings and highlighted the significance of the epigenetic regulation in plant grafting. In the revised version, we will ensure that the heading is changed. We appreciate the reviewer's feedback regarding the Conclusion section and the readability of the manuscript. We apologize for any difficulties caused by the previous version with track changes.

This manuscript is a resubmission of an earlier submission. The following is a list of the peer review reports and author responses from that submission.

Round 1

Reviewer 1 Report

Since the title refers to plants in general (Epigenetics and Plant Grafting) I suggest mentioning some research and current knowledge on perennial plants, such as fruit trees. The other option is to modify the title to meet the vegetables that are elaborated on in the paper. 

Line 190: Please provide the full Latin name for the first mention. 

There are minor typos and errors that should be checked throughout the entire manuscript, such as:

Line 412: Yet, how the process of grafting affects gene expression cannot BE ignored.

The paper should be carefully checked for the English language as well. 

The paper should be carefully checked for the English language as well. 

Reviewer 2 Report

This manuscript is an interesting overview of current information of the interrelationship of epigenetics and grafting. The flow information is well-constructed and the explanatory diagram is helpful. The references are pertinent. Avenues for future research are indicated. Publication is recommended after some minor amendments, noted below.

Line 54... ? prevent 'the' juvenile state or ? states?

Line 69 …. suggest replace 'fast' with 'quickly' since 'fast' has two meanings, either swift or to hold as in 'hold fast', i.e. hold tightly. The difference matters here so it should be clarified.

Line 82 … suggest inserting 'potentially' before 'heritable and stable modifications.....'.

Line 95-96... replace 'apparent' with 'appropriate' or 'constructive'

Lines 99-101. As currently written, this does not constitute a sentence. Please re-write.

Line 111... change 'in' to 'over'

Line 119... insert 'although before 'siRNAs'. Also, insert 'RNA-directed DNA methylation' in front of RdDM, and place RdDM in parentheses as this is the first time that term is used. It need not be repeated latter, i.e. line 277.

Line 296... change to 'pholem-associated cells …' and in Line 297, should it be 'the level of viroid?, or the 'possibility of viroid infection'?

Line 327... ? Studies that or 'Some studies'? This sentence is currently incomplete.

Line 333. Suggest ending the sentence with products, and beginning the new sentence with 'The European ….'

Line 388... capitalize the 's' in small

Line 412 insert 'be' between 'cannot ' and 'ignored'.

Line 435... insert 'an' before 'interscion' in both instances.

The quality of English is excellent in this paper.

Reviewer 3 Report

The work is quite well written - it is easy to read. Unfortunately, despite the fact that it is quite a good compendium of knowledge on the subject, I learned little new from it. The information gained is available in previous publications on the subject, e.g., in X, Y, or in basic works and even textbooks on epigenetics. In conclusion, the work is not very innovative and does not represent a significant contribution to the development of the discipline.

Reviewer 4 Report

This review seeks to provide evidence for the role of epigenetic mechanisms in the grafting process. The topic is interesting, but the form of the manuscript makes it difficult to provide an adequate assessment of its scientific merit. The authors should be more precise and the experiments of other studies must be well explained, otherwise readers will be confused.

Some suggestions follow that should be taken as an example for the whole text.

1)    Lines 84-86. “The three epigenetic modifications: DNA methylation, histone modification, and non-coding RNAs (ncRNAs), which are either small RNAs (sRNAs), or long non-coding RNAs (lncRNAs), are respon-sible for epigenetic alterations.”

This sentence is not properly correct: ncRNAs are not epigenetic modifications.

2)    Line 89-91 “Instead 89 of changes in the primary DNA sequence, epigenetics is concerned with stable and potentially heritable alterations in gene expression brought on by the way DNA is packed [19].”

This is concept is a repetition of lines 82-84

“Epigenetics is the study of heritable and stable modifications in chromatin structure that have a substantial influence on gene expression and cellular function without involving alterations to the underlying DNA sequence”

3)    Line 92-93 “ It has been associated with almost every aspect of plant development and environmental factors that plants interact with”

 Epigenetics has been….

 4)    Line 99-100 “Discovering the molecular mechanism of grafting, with an emphasis on the interactions between rootstock and scion in woody plants, and the information amassed to date about the genetic and epigenetic impacts associated with this process”

This sentence seems incomplete

5)    Line 107 “With the special emphasis on the association  between grafting and epigenetics.”

This sentence reiterates the concept explained by the previous sentence Lines 102-107

6)    Figure 1. The model does not explain nothing.

The figure legend states that "non-coding RNAs (ncRNAs), 5mC DNA, histone modifications, and m5C RNA can be transported over long distances."

Are you sure that DNA and histone modifications can be transported?

This sentence needs to be rewritten

7)    Line 199 “However, siRNAs from the scion have been proven to cause RdDM in the root-stock [22], the reciprocal event has yet to be confirmed.

I would change it in:

“However, while scion siRNAs have been shown to cause RdDM in the root strain, the reciprocal event has yet to be confirmed”.

8)    Lines 121-129. This part is very confusing.

For example,

“Plants have three fundamental epigenetic processes that may activate or silence plant genes.”

What are these 3 mechanisms?

Why do I introduce three mechanisms if the next sentence only states the association of grafting with DNA methylation and ncRNA?

9) Lines-138-141

“Some DNA methylation-related genes, such as Methyl Transferase (MET) 1, showed drastically modified expression profiles in tomato/eggplant grafted plants, albeit these profiles were reversed in progenies compared to their seed non-grafted controls [25]. It was one of the first studies to explicitly link DNA methylation and epigenetics to grafting, particularly in scions.”

This sentence is poorly written and incomplete. The study by Wu et al. shows that grafting produces heritable changes in DNA methylation and that the observed variation may be due to altered gene expression of epigenetic regulators such as met1.

As written, the sentence does not explain that a characterization of DNA methylation was performed, and it appears that conclusions were made solely on the basis of analysis of Met1 expression. Moreover, the important aspect of the study, heritability, is completely missed.

These are just a few examples but there are many other things to be fixed throughout the text